# Optical Fiber–Based Continuous Liquid Level Sensor Based on Rayleigh Backscattering

**DOI:** 10.3390/mi13040633

**Published:** 2022-04-17

**Authors:** Xingqiang Chi, Xiangjun Wang, Xuan Ke

**Affiliations:** 1College of Electrical Engineering, Naval University of Engineering, Wuhan 430033, China; chixingqiang@wtc.edu.cn (X.C.); 20180013@wtc.edu.cn (X.W.); 2School of Mechanical and Electrical Engineering, Wuhan Polytechnic, Wuhan 430074, China; 3School of Physics and Information Engineering, Jianghan University, Wuhan 430056, China

**Keywords:** optical fiber, liquid level sensor, Rayleigh backscattering, cryogenic propellant

## Abstract

This work reports an optical fiber–based continuous liquid level sensor for cryogenic propellant mass gauging, which has significant advantages over the existing liquid level sensors in terms of accuracy, simplicity, and reliability. Based on Rayleigh backscattering coherent optical frequency domain reflectometry, every point of the sensing fiber is a liquid sensor which is able to distinguish liquid and vapor. We obtained a measurement accuracy of 1 mm for the optical fiber sensor by measuring both liquid nitrogen and water levels. For the first time, for practical applications, we experimentally studied the influence of ambient temperature and strain changes on the sensing performance as well as the repeatability of the optical fiber–based liquid level sensor’s measurements.

## 1. Introduction

Liquid level sensing plays an essential role in industrial applications such as chemical processing, fuel storage and transportation systems, oil tanks/reservoirs, and wastewater treatment plants [1]. Optical fiber–based liquid level sensors have been widely researched because they can work in harsh environments with inherent advantages. They have many unique features that only optical fiber offers and that distinguish their performance from that of traditional mechanical or electronic sensors [2,3,4]; these include electromagnetic interference immunity, the use of inert materials that do not trigger explosions, remote sensing capabilities, small size, light weight, and high accuracy and resolution [5]. Several sensors are used for level measurement that are based on different technologies—for example, multi-electrode capacitance [6], ultrasound [7], radiation [8], vibration [9], fluctuation [10], radio [11], and light reflection [12]. However, due to their ability to conduct electricity, their use of electrical signals, and risk of corrosion the use of such types of sensors is not adequate for harsh environments [13]. At the moment, the most widely used optical fiber–based liquid level measurement techniques are based on long period gratings (LPG) [14,15,16], Fabry–Perot cavities [17,18], fiber Bragg gratings (FBG) [1,19], or multimode interference [20,21]. Accurate and reliable liquid level sensors are crucial for extreme applications such as cryogenic propellant mass gauging for various space missions. This is particularly true for the long-term in-orbit storage and handling of large quantities of cryogenic liquid propellants. Moreover, oil may leak from transportation pipelines on land or under sea, and hazardous liquid chemicals can spill at refinery storage facilities. It is important to rapidly detect the presence and location of any liquid spills and oil leaks in order to minimize the risk and damage caused in order to ensure environmental protection and natural resource conservation.

In this paper, we report a continuous liquid level optical fiber sensor using a sensing optical fiber cable and Rayleigh backscattering C-OFDR [22,23]. The sensor measures the liquid level continuously and is designed to replace the silicon diode–based liquid level sensing probes currently used in cryogenic propellant mass gauging systems [24,25]. Acting as an array of numerous liquid point sensors, the fiber sensor offers truly distributed measurements and is more accurate and simpler than silicon diode liquid sensors. The sensing fiber cable can be installed in a propellant tank in the same manner as a silicon diode sensing probe.

The sensing fiber cable reported in this paper can be easily manufactured for hundreds of kilometers with a low cost and can be used with Raman or Brillouin scattering-based distributed long-range sensing applications [26]. These include detecting the presence and location of hazardous liquid spills and oil leaks from transportation pipelines or refinery storage facilities. When the sensing fiber cable is exposed to a liquid spill or oil leak, the different heat capacities of the liquid or oil cause the temperature of the submerged fiber sections to vary from that of the sensing fiber sections that have not been exposed to liquid or oil. Then, the temperature difference and the event location can be determined from the changes in the optical scattering as measured by the various processes.

## 2. Materials and Methods

The optical fiber–based continuous liquid sensor simply consists of one single sensing fiber and an optical signal interrogation unit, as illustrated in Figure 1a. The operation principle of the optical fiber liquid sensor is very similar to that of silicon diode–based liquid point sensors. For a typical silicon diode sensing probe, a large number of sensing diodes are arranged at different heights across the probe, as illustrated in Figure 1b. Each diode is actively heated by feeding it a constant current. As the diodes are heated, those submerged in the liquid propellant behave differently in terms of temperature compared to those surrounded by the vapor. The temperature increase in the diodes in the liquid will be noticeably less than that in the vapor, and thus the liquid level is located at the point where a sudden change in temperature occurs between nearby diodes. Since the voltage across a diode is directly related to its temperature, the liquid level can be determined by measuring the voltage of the diodes. The temperature difference between the submerged and un-submerged silicon diode sensors is utilized to distinguish liquid from vapor. Then, the propellant liquid level can be further determined on ground or under “settled” conditions in low gravity.

The optical fiber–based liquid level sensor operates under a similar mechanism as silicon diode sensors. To determine the liquid level, the sensing fiber is actively heated and the signal interrogation unit measures the temperatures everywhere across the sensing fiber. Similar to diode sensors, the portion of the sensing fiber submerged in the liquid has a lower temperature than the portion exposed in the vapor, meaning that we can identify the liquid level by examining the temperature differences across the fiber. Optical fiber is typically coated with UV curable acrylate, polyimide, or metals. Metal coatings can be Cu, Al, Ag, Au, etc. As metal coatings on optical fiber has intrinsic resistance, when electrical current conducts along them, electrical power is transferred to heat and then increases the fiber temperature. Only metal-coated optical fibers, which are heated electrically, can raise the fiber temperature in order to distinguish the interface between liquid and vapor. We chose Cu-coated optical fiber as the sensing fiber because the Cu coating has higher resistance than the Ag or Au coating. The higher resistance the fiber coating has, the more heat is transferred to the optical fiber for the same amount of electrical current. Figure 2a illustrates the sensing fiber structure, which consists of a fiber core, fiber cladding, and Cu coating. The Cu coating thickness is ~20 µm, the fiber core diameter is ~10 µm, and the fiber cladding diameter is 125 µm. We spliced the Cu-coated fiber to a telecom optical single-mode fiber SMF28, which was connected to the signal interrogation unit located remotely. The signal interrogation unit employed C-OFDR to measure the temperature and location of “every point” of the sensing fiber submerged in liquid or exposed in vapor [26,27]. Figure 2b illustrates the Rayleigh backscattering process, where the signal interrogation unit sends an optical pump forward to generate a Rayleigh backscattering signal in the fiber core. The spectra feature of the Rayleigh signal scattering from every fiber point depends on the temperature of the fiber point. By analyzing the Rayleigh scattering signal, the signal interrogation unit was able to determine the temperature and location of each fiber point. To distinguish liquid from vapor, the optical sensing fiber was actively heated through conducting an electrical current along the Cu coating [28]. The temperature of the fiber segments submerged in liquid was limited by the boiling point of the propellant liquid, which was significantly different from the temperature of the fiber segments exposed to the propellant vapor. The optical signal interrogation unit distinguished the temperature difference between the submerged and un-submerged fiber segments and determined the exact locations of the interfaces between the liquid and vapor. Although the authors of Ref. [28] first reported the use of the liquid level optical sensor with the on-fiber heating method, the sensor’s accuracy, repeatability, and stability under the influence of ambient temperature and vibration have not been studied yet.

The signal interrogation unit we used was a Rayleigh backscattering C-OFDR made by Luna Technologies, Roanoke, VA, USA (Model # OBR4400). Unlike optical time-domain reflectometry (OTDR) using a pulsed laser source, C-OFDR uses a highly coherent continuous-wave laser source, whose wavelength is linearly swept during sensing measurements. After performing Fourier transform on the backscattering optical interference signal detected by the receiver of the interrogation unit, the frequency of the interference signal can be extracted and mapped to the location of the respective fiber segment scattering the light back. The frequency shift is proportional to external measurands affecting the fiber segment, such as temperature or strain. The spatial resolution δL of C-OFDR is determined by the following equation:(1)$$\delta L=\frac{\lambda_{S}\lambda_{E}}{2n\left(\lambda_{E}-\lambda_{S}\right)}$$
where $\lambda_{S}$ and $\lambda_{E}$ are the starting and ending wavelengths of the laser source sweeping range, and n is the effective refractive index of the fiber core. As a result, by scanning optical wavelength over the range of tens of nanometers, OFDR can nominally achieve spatial resolution in the range of tens of micron. OBR 4400 can linearly sweep the laser wavelength from 1520 nm to 1610 nm, which results into ~10-micron spatial resolution for telecom optical fiber SMF28 made by Corning Inc. of New York (n ≈ 1.46 for SMF28), in theory. Overall, the OBR 4400 demonstrated that the C-OFDR instrument can measure temperature along a 70 m long single-mode or gradient index multimode optical fiber with a 1 mm spatial resolution and 0.1-degree Celsius temperature resolution [23,29]. The 1 mm spatial resolution corresponds to 1 mm liquid level measurement accuracy. Under a continuous reading of the liquid level with a 1 mm measurement accuracy, for a 60 foot (18.3 m) tall propellant tank on ground or under “settled” low-gravity conditions, for 2% fill level, the liquid level is 1.2 feet (0.366 m) and the gauging uncertainty of the fiber sensor is calculated to be 0.27% (1 mm spatial resolution divided by 0.366 m liquid level). For 98% fill level, the liquid level is 58.8 feet (17.922 m) and the gauging uncertainty is calculated to be 0.0056% (1 mm spatial resolution divided by 17.922 m liquid level). As a comparison, silicon diode–based point liquid level sensors have been demonstrated to have a 0.5% gauging uncertainty [25]. Therefore, the measurement uncertainty of the fiber sensor varies from 0.0056% to 0.27%, which is much better than the 0.5% uncertainty of the silicon diode sensors.

Regarding the techniques used to heat the optical sensing fiber, both optical heating [30] and on-fiber electrical heating techniques have been demonstrated to be effective for liquid level sensing [28]. Optical heating is achieved by coupling a high-power laser with a high-attenuation optical fiber. The fiber core absorbs optical power and transfers much of this into heat to heat the fiber. The optical heating technique has two distinct drawbacks: (1) the length of an optically heated sensing fiber is limited to within several meters as the optical power heating the fiber decays exponentially along the fiber length. For an attenuation optical fiber with a 0.2 dB/cm optical absorption coefficient, a 1-watt optical power is injected into the fiber and the optical power is decayed to 1 × 10^−6^ W after heating a 3 m long fiber. The low optical power makes it impossible to heat the sensing fiber further; (2) the temperature profile of the optically heating fiber exponentially decays and is not uniform along the sensing fiber length. The temperature distribution in a cryogenic propellant tank is complex and often changes dynamically. Additionally, it is difficult to distinguish the temperature difference between the propellant liquid and vapor. Therefore, we chose to use the on-fiber electrical heating method to conduct liquid level measurements, where the fiber conducts electrical current to the metal coating of the optical fiber.

## 3. Experiments

We previously reported a liquid-sensing fiber structure consisting of an optical fiber and an Ni:Cr alloy resistive wire encapsulated within a thin tube [22]. The sensing fiber structure was complicated and consisted of three parts: an optical fiber, an alloy wire, and a thin tube. We used the thin tube to confine the optical fiber and the alloy wire together. Electrical current was conducted through the resistive wire in order to heat the optical fiber. However, the optical fiber was not heated uniformly, as the spacing between the optical fiber and the heating alloy wire varied inside the thin tube. To overcome the drawbacks of the sensing fiber structure, we found that the copper coating of the Cu-coated optical fiber had a sufficiently high resistance to be used as a heating source, which significantly simplified the previously reported sensing fiber structure. In this paper, we used a Cu-coated single-mode optical fiber as the sensing optical fiber to conduct liquid level measurements. The Cu coating encapsulated the entire fiber cladding and core. When electrical current was conducted along the Cu coating, the fiber core could be heated efficiently and uniformly. The Cu-coated optical fiber functioned as a Rayleigh backscattering optical fiber and a heating electrical wire of the previous sensing fiber structure.

To demonstrate that the fiber sensor could remotely distinguish the boundary between the liquid nitrogen and nitrogen vapor, we spliced the ~0.78 m long Cu-coated optical fiber to a ~64 m-long telecom fiber SMF28 with a 125 µm cladding diameter. Then, SMF28 was connected to OBR4400. The optical signal sent by OBR4400 was guided in the fiber core in order to sensing the temperature change along the fiber. A current supply was connected to the Cu coating of the sensing fiber. When the electric current flowed along the Cu coating, the optical fiber was heated. The sensing fiber was fixed on a wood stick and part of the sensing fiber was inserted into a cylindrical liquid nitrogen tank that was 1 m high and had a half a meter inside diameter. Due to the large quantity of the liquid nitrogen tank, the change in the liquid nitrogen level was negligible when heating the sensing fiber. As shown in Figure 3, the sensor clearly distinguished the boundary between the liquid nitrogen and nitrogen vapor, which was located at 64.30 m with reference to the original point of the OBR4400 unit, as indicated with the dashed green line in the figure. The sensing fiber was located between the dotted blue line and the solid red line and was heated with a 200 mA electrical current. The temperature of the fiber segment between the dotted blue line at 63.86 m and the dashed green line at 64.30 m was increased by 5 °C to 15 °C, which indicated that this part of the fiber was above the liquid nitrogen. As its temperature was limited by the boiling temperature of liquid nitrogen, the temperature of the fiber segment between the dashed green line at 64.30 m and the solid red line at 64.64 m was increased slightly by ~2 °C, which indicated that this fiber segment was submerged in liquid nitrogen. By measuring the temperature of the heated optical fiber outside and inside liquid nitrogen, the sensor was able to distinguish the boundary between the nitrogen vapor and liquid. In comparison, the silicon diode point sensor needed ~30 mA to heat itself in order to distinguish the nitrogen liquid from the vapor [25]. If we use an optical fiber with an 80 µm cladding diameter instead of the fiber with a 125 µm cladding diameter, then the heating current can be reduced to ~82 mA, which is comparable with the silicon diode liquid sensor.

To demonstrate that the fiber sensor was able to measure the liquid nitrogen level with a 1 mm spatial resolution, we mounted a ~300 mm long sensing fiber on a ruler. The ruler was then fixed vertically in an open cooler with a ~200 mm height and ~150 mm inside diameter. Liquid nitrogen contained in the cooler evaporated gradually at room temperature. The liquid nitrogen level was simultaneously measured by the fiber sensor and by the ruler, as illustrated in Figure 1a. The sensing fiber was heated with a 200 mA electrical current. The liquid nitrogen level measured by both the fiber sensor and the ruler is shown in Figure 4. The fiber sensor measured the liquid nitrogen level as accurately as the ruler with a correlation factor of 0.9982, which proves that the fiber sensor has a 1 mm spatial resolution, the same as the ruler.

It is commonly known that performance of C-OFDR-based optical fiber sensors can be influenced adversely by temperature and strain changes in ambient environments [23]. The optical fiber liquid sensor can distinguish the temperature difference between the fiber segments submerged in liquid and those outside the liquid and determine the interface between the liquid and vapor. As the sensor detects the temperature difference instead of measuring the absolute temperature along the sensing fiber, the liquid level measurement is not affected by the temperature fluctuation in ambient environments. Hence, the following experiments were focused on investigating the influence of strain change, i.e., vibration, on the sensing performance. Due to safety concerns, we used the fiber sensor to measure the water level instead of liquid nitrogen while a container was placed on a running car engine, as shown in Figure 5. We used a CSI 2120 vibration analyzer made by Emerson Process Management to the measure vibration at the container location. Figure 6 shows the vibration spectrum with a 0.477 inches/second peak velocity at a frequency of 1408.6 cycles per minute.

Firstly, we investigated the noise level of the fiber sensor without and with engine vibration. A 0.6 m long sensing fiber was spliced with a ~9 m long telecom fiber SMF28, which was connected to the OBR 4400. When the car engine did not vibrate, the maximum temperature fluctuation, i.e., the maximum noise, along the optical fiber was about ~1.0 °C, as shown by the green line in Figure 7. When the car engine vibrated, the purple line in Figure 7 shows that the maximum noise induced in the fiber was ~5.0 °C, which was repeatedly measured several times. Therefore, as long as the electrical current flowing along the metal coating raises the fiber temperature above 5.0 °C, the sensor is able to distinguish the water level. We heated the sensing fiber with a 300 mA current while the car engine was running. As shown by the red line in Figure 7, the temperature of the sensing fiber, located from 9.33 m to 9.93 m with reference to the original point of the signal interrogation unit, increased by ~9 to 21 °C. The temperature increase was significantly larger than of the noise level (5.0 °C) with the car engine vibrating. Thus, the temperature changes and locations of the sensing fiber could be accurately measured by the signal interrogation unit regardless of vibration.

Next, we used the same piece of the sensing fiber to measure the water level in the container placed on a running car engine. Different amounts of water were filled into the container. A ruler with a 1 mm accuracy was used to measure the water level when the car engine was not running. The engine was turned on. The engine vibration caused the water level up and down within a spatial range of 1~2 mm. The water level was measured by the fiber sensor with 300 mA current applied to the Cu coating. After the measurement was completed, the engine was turned off and the container was filled with more water. The ruler was used to read the water level. Then the engine was again turned on and the fiber sensor was used to repeat the measurement of the water level. We repeated the procedures 11 times to measure 11 water levels. Figure 8 shows a comparison between the results measured by the ruler and the fiber sensor. Both measurements were consistent with a correlation factor of 0.9934. This demonstrates that the fiber sensor is able to measure liquid level displacement with 1 mm accuracy, as good as the ruler.

Finally, we examined the repeatability of the fiber sensor measurement. A 0.35 m-long sensing fiber was connected to OBR4400 via a ~60 m-long telecom fiber SMF28. A sensing fiber segment that was about 0.19 m long was submerged in water while the rest remained outside of the water. Figure 9 shows 10 consecutive measurements of the temperature change along the sensing fiber made every 10 min when the fiber was heated electrically with a 400-mA current. The heated fiber was located from 61.65 m to 61.979 m away from the OBR4400. The temperature of the fiber segment outside of the water, which was located from 61.65 m to 61.788 m, was increased by ~20 °C. The temperature of the fiber segment submerged in water, which was located from 61.788 m to 61.979 m, was increased slightly by ~4 °C. For the 10 measurements, in spite of the small variation from one measurement to another, the interface between the air and water was identical at 61.788 m with reference to the original point of OBR4400. Hence, the optical fiber liquid sensor was able to determine the liquid level consistently.

## 4. Conclusions

In summary, we demonstrated that the optical fiber liquid sensor can distinguish liquid from vapor remotely at distances of up to 70 m. The novelty of this fiber sensor is to raise the fiber temperature differently between one segment submerged in liquid and another segment exposed to vapor by conducting electrical current along the metal coating of a metal-coated optical fiber. Based on the temperature difference, the fiber sensor can distinguish liquid from vapor and further determine the liquid level with 1 mm measurement accuracy. We further demonstrated that the optical fiber sensor has the same measurement accuracy regardless of the influence of temperature and strain in ambient environments. The sensing performance has a good repeatability and does not drift with time. The optical fiber liquid sensor is a truly continuous liquid level sensor with much better measurement accuracy and much simpler sensing cable structure than silicon diode liquid sensors. These advantages make the fiber sensor ready to replace the silicon diode–based liquid level sensing probe currently used in cryogenic propellent mass gauging systems.

## Figures and Tables

**Figure 1 micromachines-13-00633-f001:**
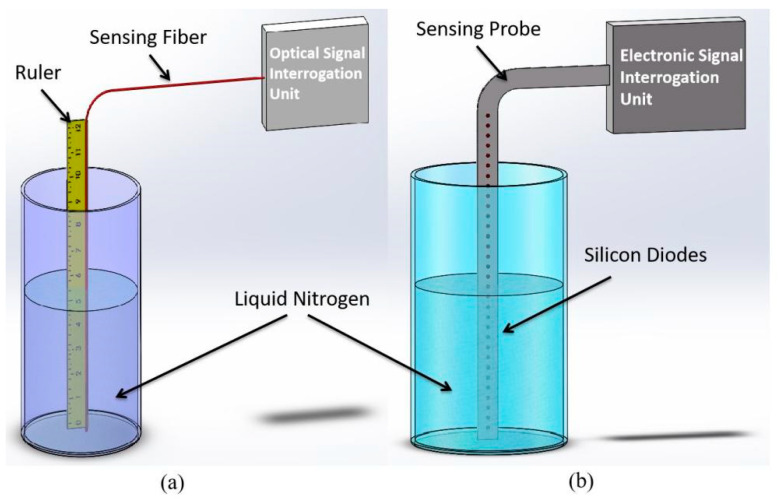
(**a**) Schematic experimental setup used for measuring liquid level using the optical fiber sensor. Liquid level was measured with a ruler and the fiber sensor (red line in the figure); (**b**) liquid level was measured with a silicon diode sensing probe, which consists of tens or hundreds of silicon diodes (black dots in the figure).

**Figure 2 micromachines-13-00633-f002:**
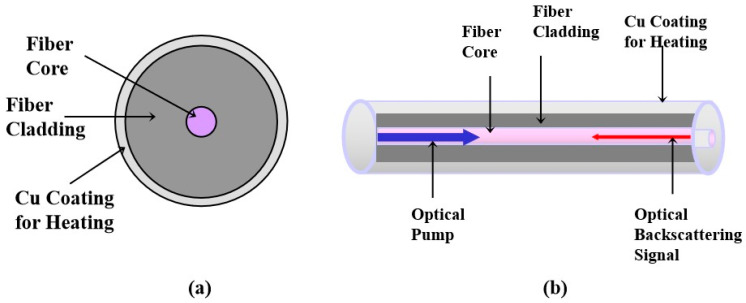
(**a**) Cross-section of the sensing fiber. Cu coating encapsulates the fiber cladding and core. (**b**) Optical Rayleigh backscattering in the optical fiber. The optical pump generates the optical backscattering signal through the Rayleigh process. The optical backscattering signal is sensitive to the fiber temperature.

**Figure 3 micromachines-13-00633-f003:**
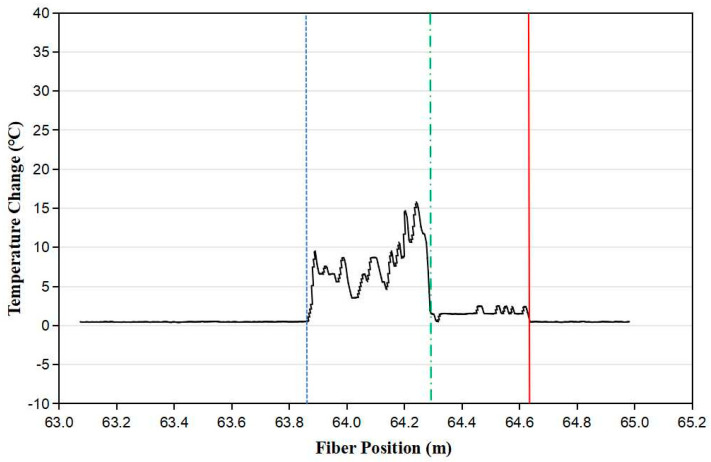
The sensor clearly distinguished the boundary between the liquid nitrogen and nitrogen vapor. The dashed green line indicates the boundary between the nitrogen vapor and liquid, which was located at ~64.30 m with reference to the original point of the OBR4400 unit. The sensing fiber was located between the dotted blue line and the solid red line.

**Figure 4 micromachines-13-00633-f004:**
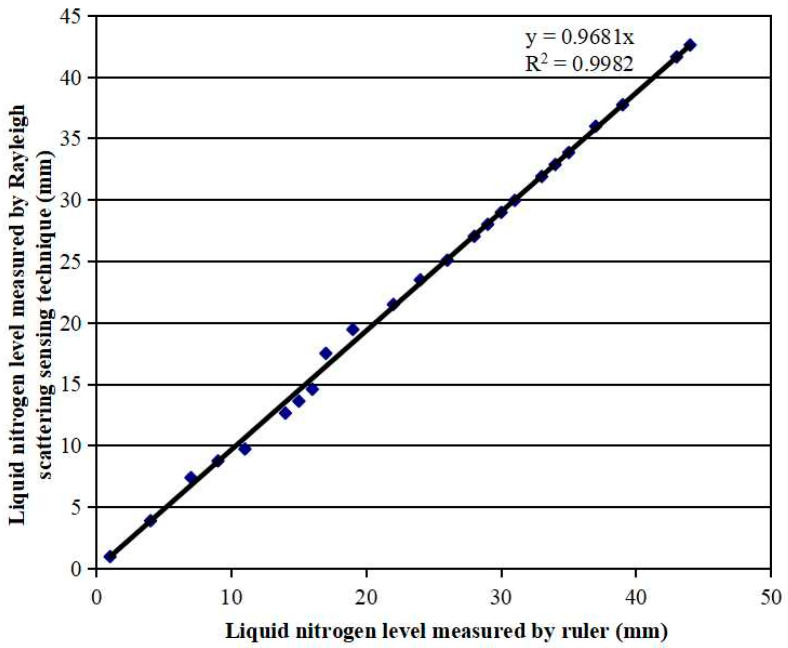
Comparison of the liquid nitrogen level measured by the fiber sensor and the ruler.

**Figure 5 micromachines-13-00633-f005:**
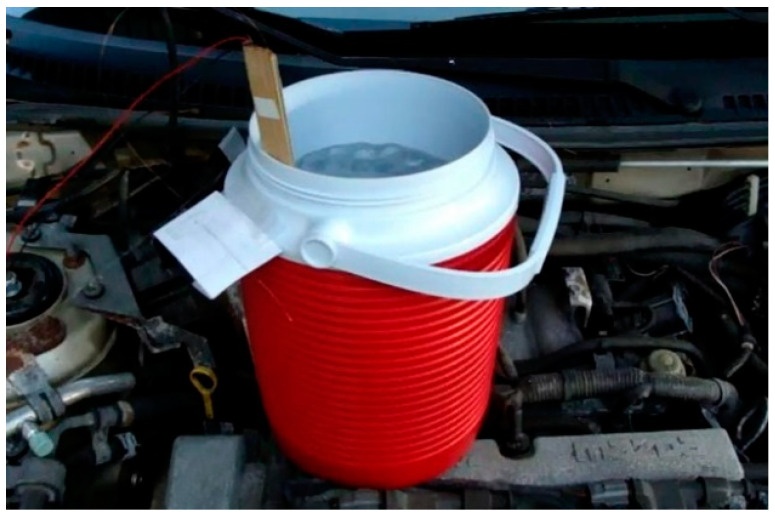
The water level measured by the fiber sensor in a container placed on a running car engine.

**Figure 6 micromachines-13-00633-f006:**
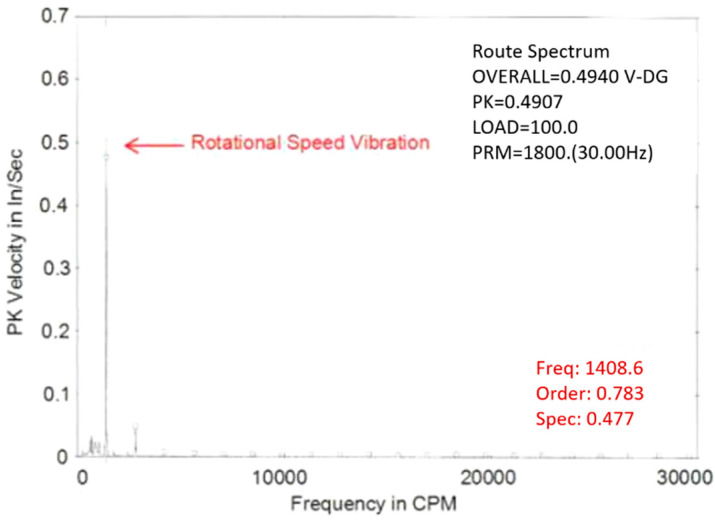
Car engine vibration spectrum at the container location with a 0.477 inches/second peak velocity at the frequency of 1408.6 cycles per minute.

**Figure 7 micromachines-13-00633-f007:**
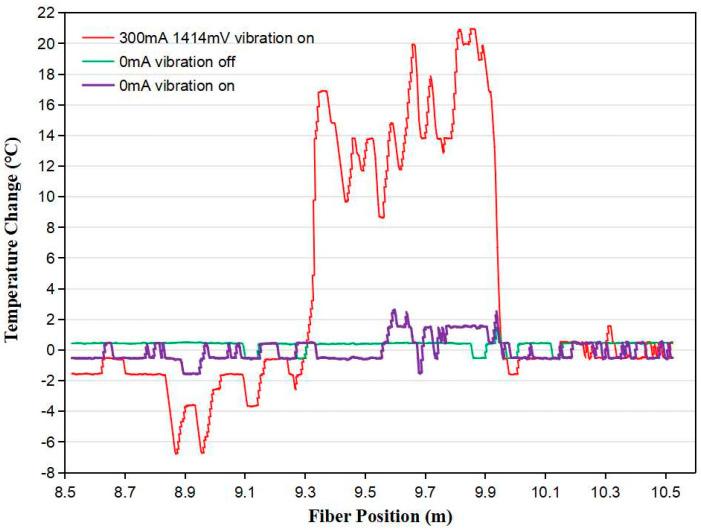
Noise—i.e., temperature fluctuation—induced in the sensing fiber without car engine vibration (green line), the noise induced with car engine vibration (purple line), and the signal strength increase caused by heating the sensing fiber (red line).

**Figure 8 micromachines-13-00633-f008:**
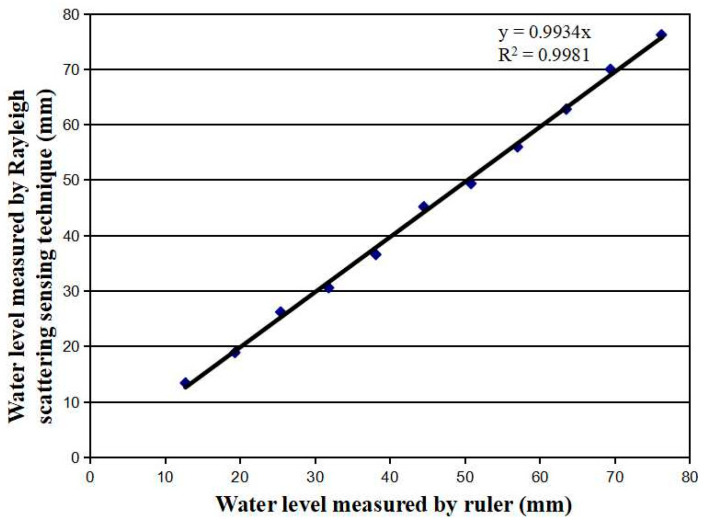
Water level was measured by means of both the fiber sensor and the ruler with a correlation factor of 0.9934, although the sensing fiber was affected by the vibration of a running car engine.

**Figure 9 micromachines-13-00633-f009:**
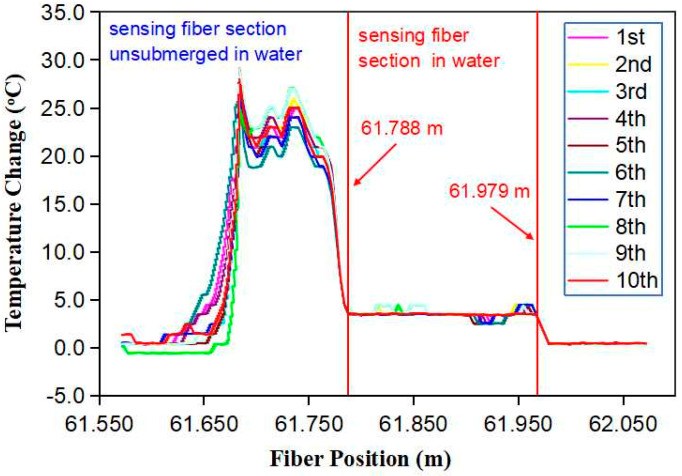
The sensor measured the interface between the water and air consecutively 10 times when the fiber was heated electrically with a 400-mA current.which are consistent.

## Data Availability

The data are contained within the article.

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
