# Peer review of "Optical Fiber–Based Continuous Liquid Level Sensor Based on Rayleigh Backscattering"

_micromachines, 2022, doi:10.3390/mi13040633_

Round 1
Reviewer 1 Report
This work presents a liquid level sensor based on Rayleigh backscattering and fabricated using a Cu-coated SMF optical fibre.
As the authors demonstrate, the system is able to measure different temperature valuess along the sensor and to be relatively independent of vibrations.
However, before considering this work for publication in this journal, the authors should highlight the advantages and novelties introduced in their sensor system. All results presented in this paper only describe the performance of the OBR4400 refractometer from LUNA technology. According to the reviewer, the results and resolutions shown here are the same as those that could be obtained with any other type of fibre optic.
For these reasons, the reviewer does not recommend publication until the authors have answered the above questions.
Author Response
Dear Reviewer,
Thanks very much for your professional comments. We responded in as much detail as possible. Please see the attachment.
Best regards
Xuan Ke

Reviewer 2 Report
The paper proposed an optical fiber continuous liquid level sensor for cryogenic propellant mass gauging based on coherent Rayleigh-backscattering optical frequency domain reflectometry, and declared a measurement accuracy of 1 mm. The idea is novel, however, authors should address the below questions before the paper can be further considered for the journal:
- Authors should provide more evidence to support the liquid level measurement accuracy of 1 mm.
- The measurement principle of the sensor needs further discussion.
- 2 is not clear and should be further explained in detail.
- In line 211, please explain what does OBR4400 mean?
- Ag has better performance in terms of thermal conductivity. Why do authors use Cu instead of Ag as the coating layer?
- The measurement uncertainty of the proposed sensor is 2% while the uncertainty of the traditional Si sensor is 0.5%. Authors need further discussion regarding the advantages of the proposed sensors over the Si sensor.
Author Response

(The authors gave the same response as above.)

Round 2
Reviewer 1 Report
After a detailed analysis of the changes introduced by the authors, I consider the article ready for publication in this journal.